# Study on Micro Production Mechanism of Corner Residual Oil after Polymer Flooding

**DOI:** 10.3390/polym14050878

**Published:** 2022-02-23

**Authors:** Xianda Sun, Mengqing Zhao, Xiaoqi Fan, Yongsheng Zhang, Chengwu Xu, Lihui Wang, Guoqiang Sang

**Affiliations:** 1Key Laboratory of “Continental Shale Oil and Gas Accumulation and Efficient Development” of Ministry of Education, Northeast Petroleum University, Daqing 163318, China; sunxianda@nepu.edu.cn (X.S.); 17863902533@163.com (X.F.); 18275580990@163.com (Y.Z.); xuchw@nepu.edu.cn (C.X.); 2Postdoctoral Research Workstation of Daqing Oilfield, Daqing 163458, China; huiaixin120710@163.com; 3Postdoctoral Research Mobile Station of Northeast Petroleum University, Daqing 163318, China; 4Exploration and Development Research Institute of Daqing Oilfield Co., Ltd., Daqing 163712, China; 5Research Institute of Petroleum Exploration and Development, CNPC, Beijing 100083, China; sgqminer@petrochina.com.cn

**Keywords:** polymer flooding, corner residual oil, microscopic visualization technology, COMSOL numerical simulation, production mechanism

## Abstract

To study the microscopic production mechanism of corner residual oil after polymer flooding, microscopic visualization oil displacement technology and COMSOL finite element numerical simulation methods were used. The influence of the viscosity and interfacial tension of the oil displacement system after polymer flooding on the movement mechanism of the corner residual oil was studied. The results show that by increasing the viscosity of the polymer, a portion of the microscopic remaining oil in the corner of the oil-wet property can be moved whereas that in the corner of the water-wet property cannot be moved at all. To move the microscopic remaining oil in the corners with water-wet properties after polymer flooding, the viscosity of the displacement fluid or the displacement speed must be increased by 100–1000 times. Decreasing the interfacial tension of the oil displacement system changed the wettability of the corner residual oil, thus increasing the wetting angle. When the interfacial tension level reached 10^−2^ mN/m, the degree of movement of the remaining oil in the corner reached a maximum. If the interfacial tension is reduced, the degree of production of the residual oil in the corner does not change significantly. The microscopic production mechanism of the corner residual oil after polymer flooding expands the scope of the displacement streamlines in the corner.

## 1. Introduction

The mechanism of polymer flooding involves adding a high-molecular-weight polymer to the injected water to increase the viscosity of the flooding phase, adjusting the water absorption profile, and increasing the swept volume of the flooding phase, thereby improving the ultimate recovery factor. Since polymer flooding in the Daqing Oilfield was put into industrial application in 1996, it has achieved remarkable technical and economic growth [1,2,3]. In 2002, the annual oil production from polymer flooding in the Daqing Oilfield exceeded 10 million tons. The tertiary oil recovery technology of the Daqing Oilfield has created a miracle in the history of world oilfield development with its large scale, high technical content, and good economic benefits [4,5,6,7]. Polymer flooding technology has become an important technical support to maintain the continuous high production and high water cut of the Daqing Oilfield and improve the oilfield development level in the later stage [8,9,10].

Polymer flooding technology has been quite mature but has encountered many problems in the application process. After injection into the oil layer, the polymer undergoes thermal degradation and further hydrolysis under high-temperature conditions, which destroys the stability of the polymer and greatly reduces its oil displacement effect [11,12,13]. At the same time, the low salinity of the formation and injection water is favorable for polymer viscosity. Because of the high salinity of the water, the viscosity of the polymer can be reduced, and the injection volume of the polymer can be increased, thereby increasing the cost, which is not conducive to the application of polymer flooding [14,15,16,17]. Therefore, it is necessary to strengthen the research on temperature resistance and salt resistance, and screen out suitable additives so that the oil displacement agent has a strong viscosity increase as well as good stability [18,19,20,21].

To study the oil displacement method for enhancing oil recovery after polymer flooding, researchers have used high-concentration polymers to increase the viscosity of the solution, or added alkali and surfactant on the basis of the polymer to form a ternary composite flooding system [22,23,24,25,26]. The oil-water interfacial tension improves the oil displacement efficiency [27].

There are obvious differences in the production mechanism of different types of microscopic remaining oil by various oil displacement technologies, and the microscopic production mechanism of the corner-shaped remaining oil after polymer flooding differs by increasing the viscosity of the displacement fluid or reducing the oil-water interfacial tension [28]. Most of the existing research results focus on the oil displacement mechanism of oil displacement technology and only analyze the macroscopic oil displacement mechanism of oil displacement agents or systems; there are few studies on a single type of microscopic residual oil production mechanism [29]. As a result, the oil displacement method does not clarify the microscopic production mechanism of the remaining oil retained after polymer flooding and cannot provide theoretical and technical guidance for the efficient development of oil reservoirs after polymer flooding [30]. According to the occurrence state and distribution characteristics of the microscopic residual oil, the finite element numerical simulation software COMSOL can establish a three-dimensional model [31]. By defining the fluid properties, simulating the seepage characteristics of multiphase fluids in porous media plays a crucial role in analyzing the microscopic residual oil production mechanism [32]. Combining COMSOL numerical simulations and microscopic visualizations of oil displacement, physical simulations can be used to comprehensively study the oil displacement mechanism of corner-shaped microscopic residual oil [33].

The microscopic residual oil production mechanism after polymer flooding is more complicated; in particular, the changes in viscosity and interfacial tension of the flooding system have a remarkable influence on the corner-shaped microscopic residual oil. In this study, microscopic visualization flooding technology was used, and the corner-shaped microscopic residual oil characteristic model was used to analyze the influence of viscosity, wettability, and interfacial tension of the flooding system on the corner-producing mechanism after polymer flooding. Based on COMSOL numerical simulation software, a model was established according to the distribution characteristics of the corner-shaped remaining oil in the microscopic visualization image, and different displacement conditions were simulated. By combining the dynamic images of the microscopic visualization of oil displacement with the numerical simulation results, the microscopic production mechanism of corner-shaped microscopic residual oil after polymer flooding was analyzed.

## 2. Experimental Methodology

### 2.1. Materials

Chemicals: Partially hydrolyzed polyacrylamide amide: molecular weight 12 million, 25 million, effective content 90%; surface active agent: petroleum sulfonate, effective content 40%; alkali: sodium carbonate, effective content 99%; the above chemical agents were obtained from Daqing Refinery Chemical Company, Daqing, China.

Oil and water: Simulated formation water; the NaCl content of clean water is 950 mg/L, and the NaCl content of sewage is 4500 mg/L. All systems use clean water to prepare the mother liquor, sewage dilutes the target liquid, and the visual model displacement experiment uses sewage. Experimental oil: Simulated oil prepared in a certain ratio of crude oil and kerosene, with a viscosity of 10 mPa·s at 45 °C.

### 2.2. Microscopic Visualization Model

Photochemical etching technology was used to create a microscopic visual glass model, in which the real pore structure photo of the cast sheet was placed on the glass coated with photosensitive material, and the contour pattern of the pore structure was copied on the glass after exposure and development. The exposed glass template was then treated with hydrofluoric acid to obtain an impression of the pore structure, and a cover plate was added for high-temperature sintering, as shown in Figure 1.

### 2.3. Interfacial Tension Test

Interfacial tension (IFT) was measured using a Grace M6500 rotating drop interfacial tension meter, Produced in the United States, using polymers with molecular weights of 25 million, petroleum sulfonate, and sodium carbonate to prepare three IFT grades of 10^−1^, 10^−2^, and 10^−3^ mN/m. Meta-composite system. The interfacial tension between the oil phase and oil displacement system was measured under a simulated Daqing reservoir temperature of 45 °C. All measured speeds were controlled at 3000 rpm. The interfacial tension value at the end of each test for 30 min [34].

### 2.4. Polymer Viscosity Test

The viscosity of the polymer was tested using a DV-II Pro viscometer, Produced in the United States. The viscosity of the solution used for polymer flooding, prepared with a polymer with a molecular weight of 12 million, was 40 mPa·s. The solution used after polymer flooding was prepared with polymers with molecular weights of 25 million and with polymer concentrations of 1500, 2000, and 2500 mg/L. A number 0 rotor was selected at six revolutions per minute; each solution was tested twice, and the average of the test results was taken to reduce the error [35].

### 2.5. Microscopic Visualization Flooding Experiment

First, the microscopic visualization model was vacuumed, and then, the oil was saturated. The difference in mass before and after saturation was used to calculate the pore volume. Further, we proceeded with water flooding at a displacement rate of 0.03 mL/h and continued water flooding until no oil was produced at the production end of the model. The polymer solution (molecular weight: 12 million, viscosity: 40 mPa·s) was driven until no oil was produced at the production end of the model. After polymer flooding, we injected high-concentration polymers (molecular weight: 25 million, 1500 mg/L, 2000 mg/L, 2500 mg/L) or polymers with different interfacial tension levels (10^−1^ mN/m, 10^−2^ mN/m, 10^−3^ mN/m) of the ASP system, until the model produced no oil. This marked the end of the experiment.The micro oil displacement system is shown in Figure 2.

### 2.6. COMSOL Numerical Simulation

The corner residual oil was mainly contained in a U-shaped structure formed by a complex pore structure. One side was in contact with the angle depression formed by a part of the reservoir, and the other side was in contact with the open space. During water flooding, owing to insufficient development of the displacement streamline, it was difficult to displace the remaining oil in the corners. After water flooding, the remaining oil was isolated in the form of oil droplets, and it remained in the dead corners of the pores that could not be swept by the injected water, as shown in Figure 3a. Polymer flooding increased the viscosity of the displacement phase. The residual oil in the corner was pulled and stripped using the polymer solution, and part of it was displaced; however, a large part of the remaining oil remained in the corner of the pore and could not be used (Figure 3b).

Based on the occurrence and distribution characteristics of corner residual oil after polymer flooding, COMSOL(COMSOL Inc., Stockholm, Sweden) finite element numerical simulation software was used to establish the corner residual oil model, as shown in Figure 4. The corner-shaped model is represented on the micrometer scale, with the entrance on the left and exit on the right. By defining the boundary and solving the domain, the mesh is divided into triangles, and ultra-refinement is performed when constructing the division element to ensure that the simulation calculation process always converges. The two-phase fluids were displacement fluid and oil. The viscosity of the oil was set to 10 mPa·s. The displacement phase parameters were set based on the simulation conditions. Microscopic corner residual oil has oil-wetting properties. This model was used to simulate the different displacement conditions. A numerical simulation was used to study the microscopic production mechanism of corner residual oil after polymer flooding [36,37,38].

## 3. Results and Discussion

### 3.1. Microscopic Visualization Flooding Experiment

#### 3.1.1. Effect of Viscosity on the Production of Corner Residual Oil

Polymer solutions of different concentrations were prepared using a DV-II Pro viscometer to test the polymer viscosity. After water flooding, polymers with a molecular weight of 12 million were used to prepare a solution with a viscosity of 40 mPa·s for displacement. After polymer flooding, polymers with a molecular weight of 2500 were used to prepare solutions with concentrations of 1500, 2000, and 2500 mg/L for displacement. The viscosity relations of polymer solutions with different molecular weights and concentrations are listed in Table 1.

Corner residual oil is mainly formed due to the insufficient development of the displacement streamline. To increase the degree of production of this type of residual oil, it is necessary to increase the sweep range of the displacement streamline of the oil displacement system. The viscosity of the displacement phase can increase the spread of the solution during the displacement process, but the production of corner residual oil is remarkably affected by the wettability. As shown in Figure 5, in the corner-shaped pores with water-wet properties, the microscopic corner residual oil is not used even when the viscosity of the polymer continues to increase. Corner residual oil subjected to stress analysis can explain this phenomenon.

Figure 6 shows the force analysis of corner residual oil with water-wet properties. In the displacement direction, when the polymer solution passes through the corner residual oil, the force on the remaining oil droplets includes the carrying force of the polymer to the crude oil F_1_, cohesive force F_2_ that prevents the remaining oil from peeling off, and remaining oil rotation under the action of an external force. The centrifugal force is F_3_. If the corner residual oil moves, the displacement power must be greater than the displacement resistance.

For the remaining oil to flow out of the corner or for small oil droplets to break out under the action of the polymer solution to reduce the volume, the following two conditions must be satisfied [39]: the carrying force of the polymer on the crude oil must be greater than the cohesive force preventing the remaining oil from peeling off, and the centrifugal force of the crude oil rotation must be greater than the cohesive force that prevents the crude oil from peeling off.
(1)$$F_{1}>F_{2}\cos\theta$$
(2)$$F_{3}>F_{2}\sin\theta$$
where *θ* is the angle between the carrying force and cohesive force, °; and *F*_l_, *F*_2_, and *F*_3_ are the acting forces, N.

For an equal-diameter capillary with a radius *R*, the shear force acting on the wall by the displacement fluid is
(3)$$\tau =\frac{\Delta pR}{2L}$$
(4)$$\Delta p=2\mu L\frac{v^{n}}{R^{n+1}}{\left(\frac{3n+1}{n}\right)}^{n}$$

If the top of the corner residual oil is in level with the seepage wall, the carrying force *F*_1_ acting on the remaining oil during the displacement process is
(5)$$F_{1}=\tau A_{1}$$

Substituting Equations (3) and (4) into Equation (5), we obtain
(6)$$F_{1}=\frac{\mu v^{n}}{R^{n}}{\left(\frac{3n+1}{n}\right)}^{n}A_{1}$$
where *v* is the average flow rate, m/s; *R* is the capillary radius, μm; *μ* is the viscosity of the displacement fluid, mPa·s; *n* is the fluid power law exponent, *A*_1_ is the contact area between the polymer solution and remaining oil droplets, μm^2^.

If the corner residual oil deforms under the action of the carrying force, and if small oil droplets are broken, the force of the displacement fluid must overcome the cohesive force generated by the remaining oil to maintain its own shape.
(7)$$F_{2}=\frac{2\sigma }{R_{2}}A_{2}$$
where *σ* is the interfacial tension of the oil and polymer solution, mN/m; *R*_2_ is the radius of the new droplet, μm; and *A*_2_ is the external surface area of small oil droplets, μm^2^.

The ratio of these two forces is
(8)$$\frac{F_{1}}{F_{2}\cos\theta }=\frac{\frac{\mu v^{n}}{R^{n}}{\left(\frac{3n+1}{n}\right)}^{n}A_{1}}{\frac{2\sigma }{R_{2}}A_{2}\cos\theta }={\left(\frac{3n+1}{n}\right)}^{n}\frac{nv^{n}}{2\sigma }\frac{R_{2}}{R^{n}}\frac{A_{1}}{A_{2}}\frac{1}{\cos\theta }$$

When *n* = 1, the above formula is simplified to
(9)$$\frac{F_{1}}{F_{2}}=\frac{\frac{4\mu v}{R}A_{1}}{\frac{2\sigma }{R_{2}}A_{2}}=\frac{2\mu v}{\sigma }\frac{R_{2}}{R}\frac{A_{1}}{A_{2}}$$

The actual calculation was performed according to the above formula (9). It is assumed that the contact area between the polymer solution and the remaining oil droplets is the same as the outer surface area of the new oil droplets broken out, that is, *A*_1_ = *A*_2_. According to the actual parameter settings in the microscopic visualization flooding experiment, the interfacial tension (*σ* = 30 mN/m), flow velocity of the polymer solution (*v* = 10^−3^~10^−5^ m/s), and viscosity of the polymer solution (*μ* = 40 mPa·s) gives *F*_1_/*F*_2_ = (10^−2^~10^−4^) after the calculation. The results from the calculation show that if the corner residual oil is used, under the condition that the cohesion of the crude oil remains unchanged, the displacement fluid needs to produce 100–1000 times the carrying force (displacement fluid viscosity or seepage velocity) during the displacement process. An increase of 100–1000 times can produce corner residual oil with water-wetting properties, but in an actual production process, it is almost impossible to achieve this condition. The theoretical calculation of the model was used to verify the conclusion that it is difficult to produce corner residual oil. Therefore, under the condition of ensuring the viscosity of the oil displacement system, the interfacial tension of the oil displacement system can be reduced, and the wettability of the crude oil can be changed during the displacement process to increase the degree of corner residual oil production.

#### 3.1.2. Effect of Wettability on the Production of Corner Residual Oil

A corner-shaped microscopic model was used to analyze the dynamic images of the remaining oil migration with different wettability corners, as shown in Figure 7. Under the same viscosity conditions, the remaining oil in the water-wet pores was not used, whereas the remaining oil in the oil-wet pores was significantly reduced. As the viscosity of the polymer increased, the sweep range of the oil displacement system in the oil-wet pores and the degree of remaining oil production increased. The remaining oil in the water-wet pores did not move even when the viscosity continued to increase because it did not reach the utilization conditions required in the above calculations. To wet the remaining oil in the corner with water, it is necessary to reduce the oil-water interfacial tension and change the wettability before it can be used passively. This phenomenon further verifies the results of the theoretical calculations.

#### 3.1.3. Effect of Interfacial Tension on the Production of Corner Residual Oil

An ASP system with 25 million molecular weight polymers and different interfacial tension levels was used. The polymer concentration was 2200 mg/L, and the mass concentration of sodium carbonate was 1.2%. The interface of the ternary composite system was adjusted by changing the mass concentration of petroleum sulfonate. The interfacial tension was tested using a Grace M6500 rotating droplet interfacial tension meter. The test results are shown in Table 2.

After polymer flooding, an ASP system with different interfacial tensions was used to conduct microscopic visualization flooding experiments, and dynamic images of the same corner residual oil were continuously tracked. The changes in the corner residual oil after polymer flooding and ASP flooding were compared and analyzed, as shown in Figure 8. Compared with the remaining oil after polymer flooding, a decrease in the interfacial tension results in significantly reduced corner residual oil, indicating that reducing the interfacial tension of the oil displacement system can effectively produce corner residual oil.

### 3.2. Numerical Simulation of Corner Residual Oil

#### 3.2.1. Numerical Simulation of Different Viscosities

To study the influence of the viscosity change of the displacement phase on the initial migration of the remaining oil in the corner, numerical simulations under different viscosity conditions were performed [40,41,42]. By using outputting parameters such as oil saturation, velocity, and pressure in the corner during the displacement process of the same injection volume (0.2 PV), the production mechanism of the remaining oil in the corner was analyzed (Figure 9). As the viscosity increased, the remaining oil saturation and the velocity contour in the seepage direction in the corner gradually decreased. The displacement streamlines were fully developed; the depth of the displacement fluid into the corners increased, and the sweep range increased. The pressure contours at the outlet end gradually became denser; the displacement pressure difference increased, and the degree of production of corner residual oil increased. However, when the displacement time was long enough and the displacement phase viscosity reached 215 mPa·s, the remaining oil saturation in the corner was still high, indicating that it is more difficult to produce corner residual oil by increasing the viscosity of the displacement fluid. This is consistent with the results of the microscopic visualization flooding experiment. To increase the degree of microscopic remaining oil production in the corner, under the premise of ensuring the viscosity of the displacement phase, the interfacial tension of the displacement phase can be reduced, and the wettability can be changed.

The curve of the corner residual oil saturation with time under different viscosity conditions in the simulation process is shown in Figure 10. According to the calculation of the ratio of the corner residual oil to the entire model, as the viscosity of the displacement phase increases, the corner residual oil saturation gradually decreases. When the viscosity values were 40 and 215 mPa·s, the oil saturation was reduced by 1.37 and 2.87%, respectively. The numerical simulation results show that by increasing the viscosity of the displacement phase, the degree of production of the corner residual oil can be increased. The degree of production is low, however, and a large part of the remaining oil is stagnated in the corner and cannot be used.

From the formation mechanism of the corner residual oil, it can be seen that the corner residual oil is retained in the percolation process, owing to the insufficient development of the displacement streamline. By increasing the viscosity of the displacement phase, the spreading range of the displacement streamline in the corner was expanded, and the development of the displacement streamline in the corner under different viscosity conditions at the same displacement time (PV) was obtained (Figure 11). When the viscosity was 70 mPa·s, the front edge of the displacement streamline had a relatively small spread in the corner [43,44,45]. As the viscosity of the displacement phase increased, the flow cross-section of the displacement phase changed during the flow of the shrinkage channel. A pure viscous fluid relies on the inertial effect to expand the scope of the displacement streamline in the corner. The depth of the displacement phase solution deep into the corner model increased, which increased the utilization of corner residual oil.

#### 3.2.2. Numerical Simulation of Different Interfacial Tensions

The changes in the remaining oil wetting angle in the corner under the same displacement time (0.2 PV) and different interfacial tension conditions were compared and analyzed (Figure 12).

Under the same conditions of displacement phase viscosity, as the interfacial tension decreases, the oil-water interface deformation amplitude in the oil-wet pore channel and the wetting angle increases [46,47,48]. The oil saturation cloud diagram shows that the remaining oil in the corners varies as the interfacial tension decreases and that the continuous oil phase migrates to the outlet end, increasing the degree of production of the corner residual oil.

By changing the interfacial tension of the displacement phase, the production of the corner residual oil was analyzed, and the curve of the oil saturation in the corner with the injection volume under different interfacial tension conditions was drawn, as shown in Figure 13. When the polymer flooding viscosity was 40 mPa·s, the oil saturation in the corner was reduced by 1.37%. Under the same viscosity condition, when the interfacial tension was reduced to 0.3 mN/m, the oil saturation in the corner was reduced by 3.45%. When the viscosity was 215 mPa·s, the oil saturation was only reduced by 2.87%, indicating that reducing the interfacial tension can significantly increase the degree of production of corner residual oil. When the interfacial tension was reduced to 0.03 mN/m, the oil saturation in the corner was reduced by 4.51%, which was 1.06% higher than when the interfacial tension was 0.3 mN/m. When the interfacial tension was reduced to 0.003 mN/m, the oil saturation in the corner was reduced by 5.03%, which was 0.52% higher than when the interfacial tension was 0.03 mN/m. The numerical simulation results showed that when the interfacial tension level reached 10^−2^ mN/m, the degree of production of the corner residual oil was the highest. Under these conditions, the recovery rate of the remaining oil in the corner reached its limit, and the degree of production of the remaining oil increased very little, by continuing to reduce the interfacial tension. This conclusion can be used to guide field applications in conjunction with microscopic visual oil-displacement experimental data.

The influence of interfacial tension on the production of the remaining oil in the corner was analyzed according to the changes in the velocity and pressure contours in the corner during the numerical simulation, as shown in Figure 14. As the interfacial tension of the displacement phase decreases, the wetting angle of the remaining oil in the corner gradually increases during the displacement process [49,50,51,52]. The velocity contour gradually increases in the corner, the depth into the corner increases, and the remaining oil in the diagonal corner increases. The displacement fluid carries the remaining oil and migrates to the outlet end, the displacement pressure gradually increases, and the pressure contour slopes toward the outlet end. The closer it is to the outlet, the denser is the pressure contour.

From the formation mechanism of the corner residual oil, the remaining oil in the corner is retained in the percolation process, due to the insufficient development of the displacement streamline. By reducing the interfacial tension and increasing the sweeping range of the displacement streamline in the corner, the same displacement time (PV number) and the development of the displacement streamline in the corner under different interfacial tension conditions can be achieved (Figure 15). When the interfacial tension is 0.3 mN/m, the sweep range of the displacement streamline in the corner is greater than that under the condition of a viscosity of 215 mPa·s. As the interfacial tension decreases, the displacement streamlines fully develop in the corner, and the spread in the corner gradually increases, thus increasing the degree of production of corner residual oil. When the interfacial tension is reduced to 0.003 mN/m, the displacement streamline of the displacement phase in the corner model spreads to the bottom of the model.

## 4. Discussion

The conclusions of this study are listed as follows:

(1) For the corner residual oil retained after polymer flooding, the method of increasing the viscosity of the solution with a high-concentration polymer solution cannot move the remaining oil in the water-wet pores, while part of the oil-wet pores can be moved.

(2) According to theoretical calculations, if the corner residual oil of the water-wet properties is to be produced, the cohesive force of the crude oil needs to be unchanged, and the displacement fluid needs to generate 100–1000 times the carrying capacity during the displacement process. That is, the displacement liquid viscosity or seepage velocity increases 100–1000 times, which is difficult to achieve in the actual development process.

(3) The numerical simulation results show that when the interfacial tension level reaches 10^−2^ mN/m, the wetting angle changes the most; the remaining oil recovery in the corner reaches a maximum, and the oil saturation changes when the interfacial tension continues to decrease. Considering the development benefits, the interfacial tension level reaches 10^−2^ mN/m as a critical condition.

(4) The corner residual oil retained after polymer flooding is mainly due to the insufficient development of displacement streamlines. Increasing the viscosity of the displacement phase has little effect on the corner residual oil, reducing the interfacial tension and changing wettability and displacement. An alternate streamline develops at the bottom of the corner.

## Figures and Tables

**Figure 1 polymers-14-00878-f001:**
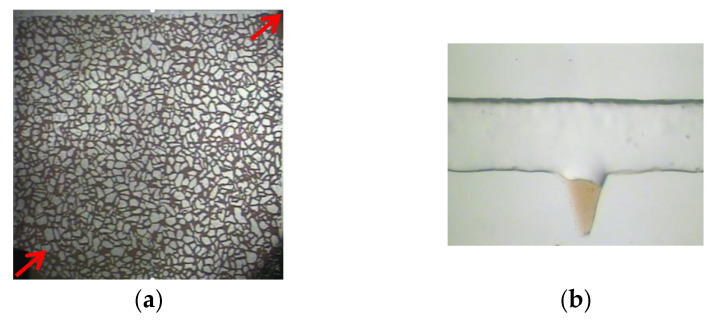
Microscopic oil displacement experimental device. (**a**) Microscopic visualization model. (**b**) Corner residual oil model. The red arrow represents the displacement direction.

**Figure 2 polymers-14-00878-f002:**
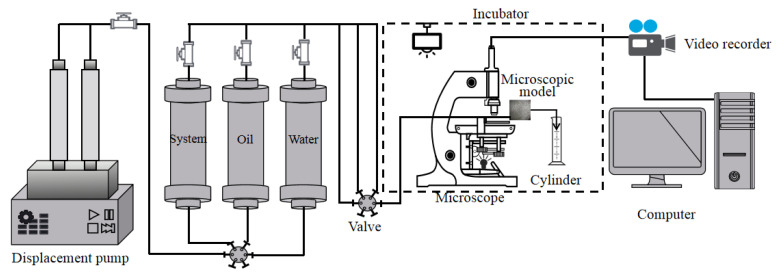
Microscopic oil displacement experimental device.

**Figure 3 polymers-14-00878-f003:**
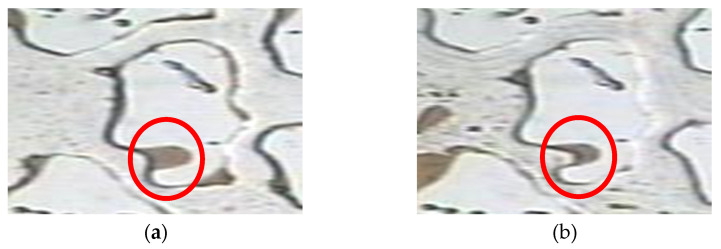
Microscopic images of corner residual oil. (**a**) After water flooding. (**b**) After polymer flooding. The corner shaped residual oil is marked in the red circle.

**Figure 4 polymers-14-00878-f004:**
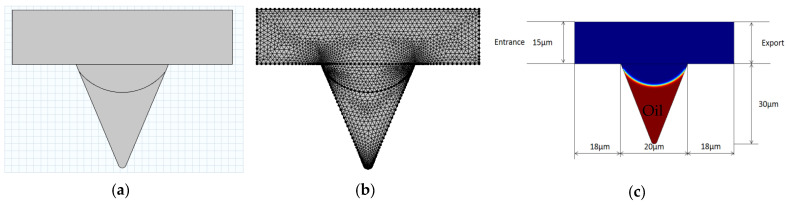
Corner model establishment method. (**a**) Build model. (**b**) Mesh division. (**c**) Defined fluid.

**Figure 5 polymers-14-00878-f005:**
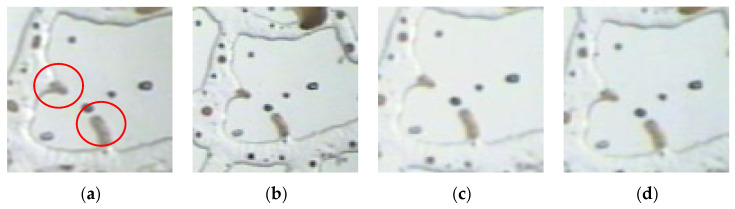
Corner residual oil production under different viscosity conditions. (**a**) Viscosity 40 mPa·s. (**b**) Viscosity 70 mPa·s. (**c**) Viscosity 125 mPa·s. (**d**) Viscosity 215 mPa·s. The corner shaped residual oil is marked in the red circle.

**Figure 6 polymers-14-00878-f006:**
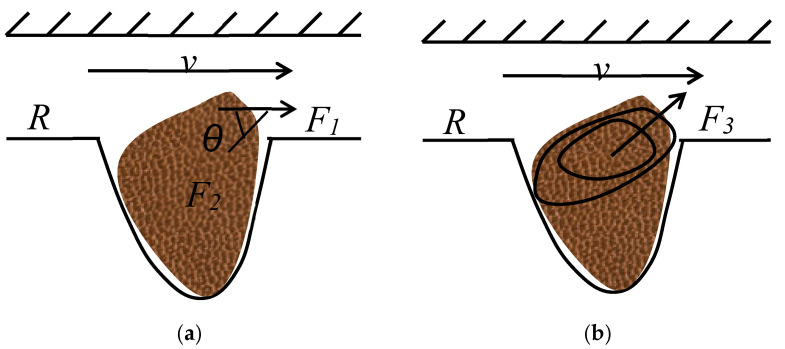
Stress analysis of corner residual oil. (**a**) Shear force effect. (**b**) Centrifugal force effect.

**Figure 7 polymers-14-00878-f007:**
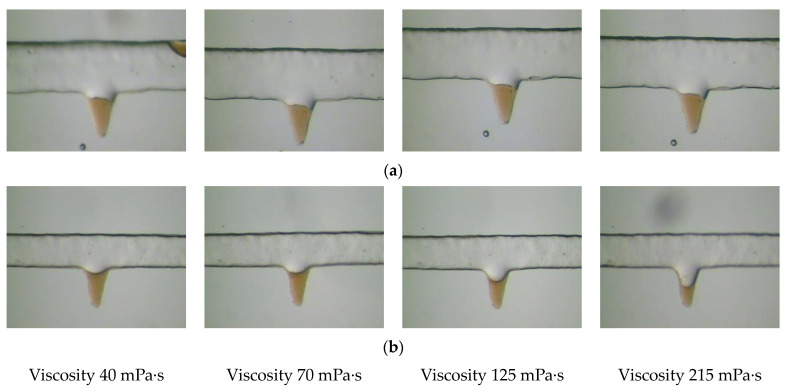
Corner residual oil production under different viscosity conditions. (**a**) Water-wet pores. (**b**) Oil-wet pores.

**Figure 8 polymers-14-00878-f008:**
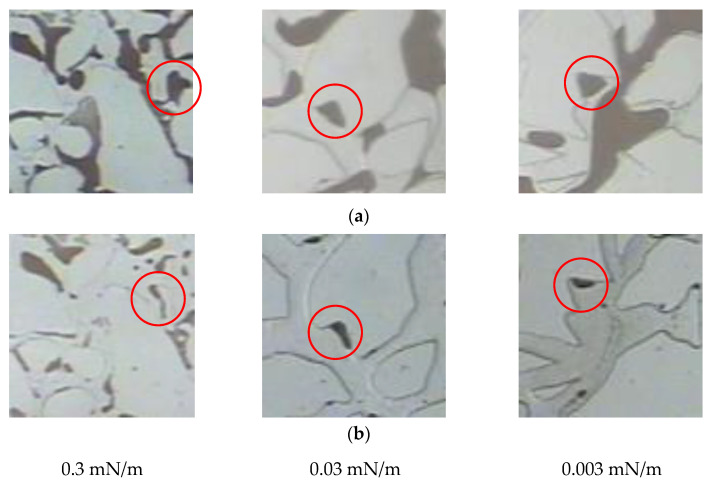
Development of corner residual oil under different interfacial tension. (**a**) After polymer flooding. (**b**) ASP flooding with different interfacial tension. The corner shaped residual oil is marked in the red circle.

**Figure 9 polymers-14-00878-f009:**
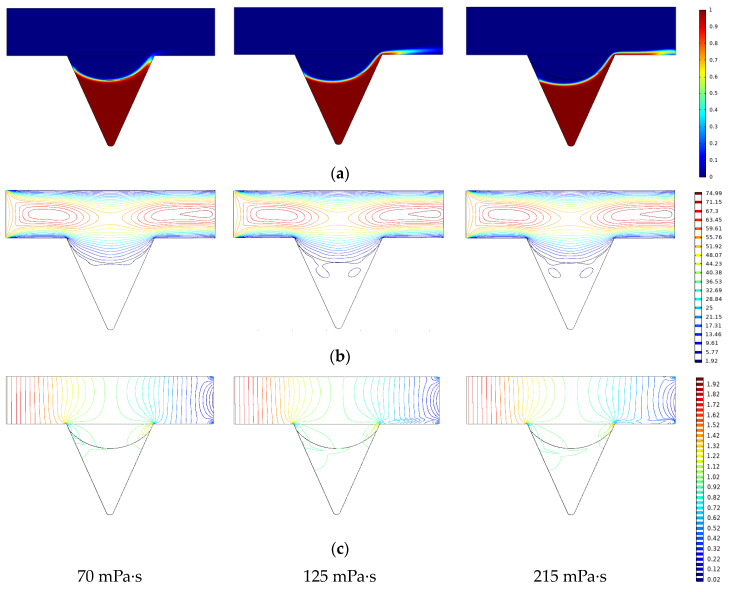
Effect of viscosity corner residual oil production. (**a**) Oil saturation. (**b**) Speed contour. (**c**) Pressure contour.

**Figure 10 polymers-14-00878-f010:**
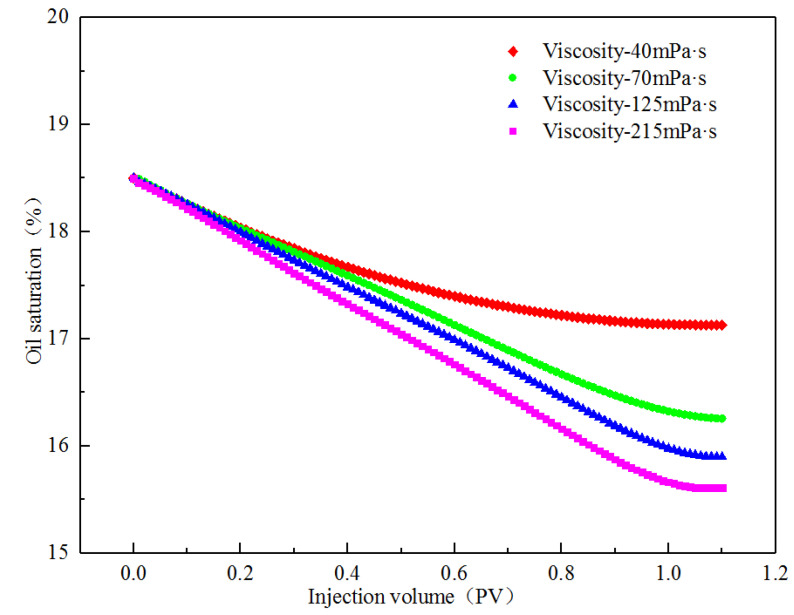
Saturation variation of corner residual oil under different viscosity conditions.

**Figure 11 polymers-14-00878-f011:**
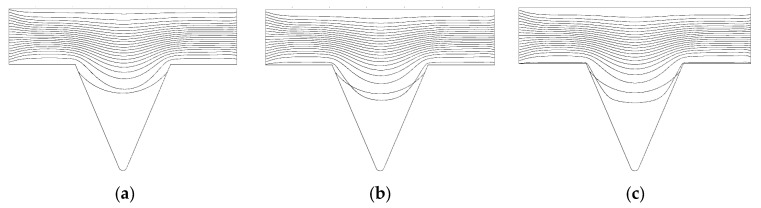
Flow line diagram in corner under different viscosity conditions. (**a**) 70 mPa·s. (**b**) 125 mPa·s. (**c**) 215 mPa·s.

**Figure 12 polymers-14-00878-f012:**
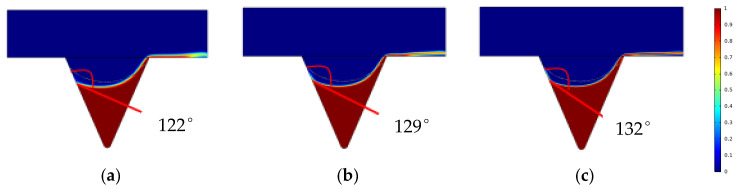
Variation of wetting angle under different interfacial tension conditions. (**a**) Interfacial tension 0.3 mN/m. (**b**) Interfacial tension 0.03 mN/m. (**c**) Interfacial tension 0.003 mN/m.

**Figure 13 polymers-14-00878-f013:**
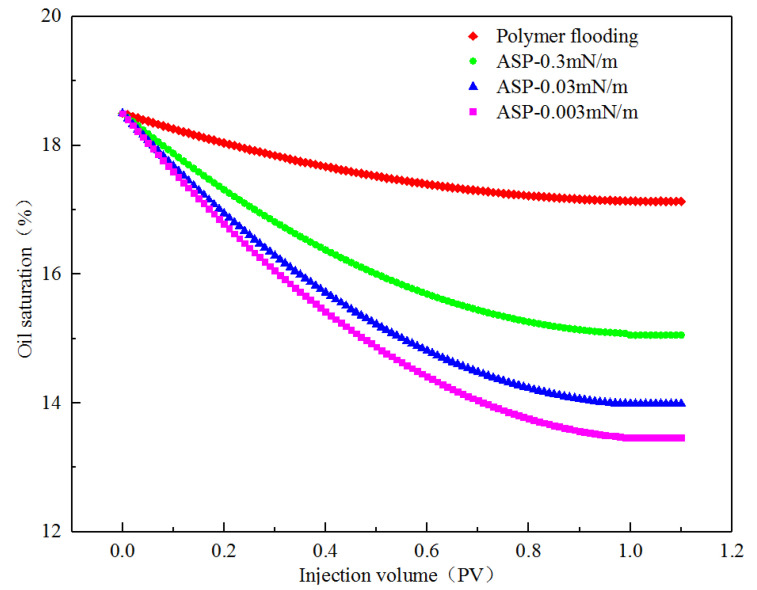
Variation of corner oil saturation under different interfacial tension.

**Figure 14 polymers-14-00878-f014:**
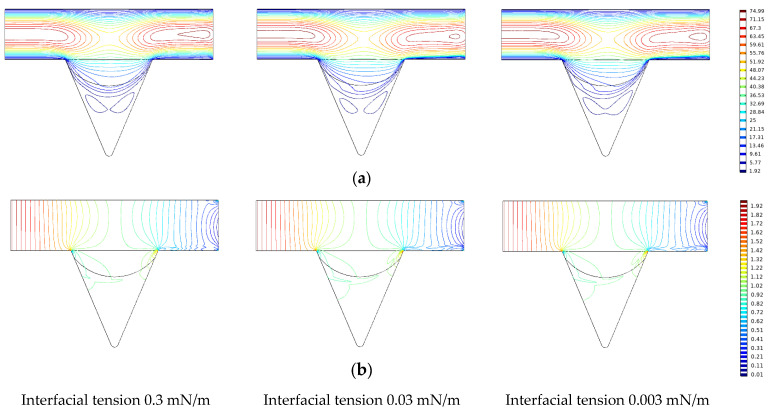
The change of velocity and pressure under different interfacial tension conditions. (**a**) Speed contour. (**b**) Pressure contour.

**Figure 15 polymers-14-00878-f015:**
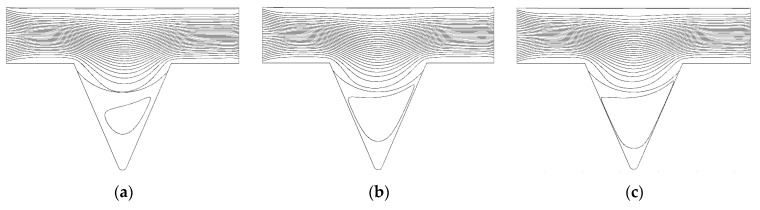
Flow line diagram in corner under different interfacial tension conditions. (**a**) Interfacial tension 0.3 mN/m. (**b**) Interfacial tension 0.03 mN/m. (**c**) Interfacial tension 0.003 mN/m.

**Table 1 polymers-14-00878-t001:** Viscosity of polymer solutions with different molecular weights and concentrations.

Polymer Molecular Weight (Ten Thousand)	Polymer Concentration (mg/L)	Viscosity (mPa·s)
1200	870	40
2500	1500	70
2500	2000	125
2500	2500	215

**Table 2 polymers-14-00878-t002:** Concentrations of ternary composite systems with different interfacial tension levels.

Petroleum Sulfonate Mass Concentration (%)	Sodium Carbonate Mass Concentration (%)	Polymer Concentration (mg/L)	Interfacial Tension (mN/m)
0.00	1.2	2200	0.3
0.02	1.2	2200	0.03
0.30	1.2	2200	0.003

## Data Availability

Not applicable.

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
