# Peer review of "Study on Micro Production Mechanism of Corner Residual Oil after Polymer Flooding"

_polymers, 2022, doi:10.3390/polym14050878_

Round 1
Reviewer 1 Report
The paper entitled “Study on micro production mechanism of corner residual oil after polymer flooding” by Sun et. al. reprts a study of the production mechanism of corner residual oil upon polymer flooding. In this study, authors used microscopic visualization technology and COMSOL simulation methods.
The manuscript has been Witten in a technical report style rather than a scientific article in a peered review journal. Therefore, Manuscript need to be revised thoroughly before resubmission to Polymer.
- Sentences used in Abstract are long and need paraphrasing to be enhance readability. All long sentences in the manuscript need to be revised accordingly.
- Abbreviations need to be defined once they appear in the text (Such as ASP,..)
- Line 40, page 1: there is a very short incomplete sentence “. Degree control ability.” What does it mean?
- Although the corner residual oil is the main keyword of the manuscript and the title, Authors did not highlight it in the purpose of the manuscript as described in the last paragraph of the introduction or any part in the introduction.
- COMSOL simulation method and its importance and benefits to the research work have not been discussed clearly in the introduction.
- Description of the Corner Residual Oil and Polymer Flooding concepts is needed to highlight the importance of the manuscript and educate non-expert readers.
- Discussion section “section 4” is very short to be called discussion. This discussion points should be introduce at the beginning to show the importance of the selected methods in the oil recovery.
Therefore, the manuscript as such is not acceptable for publication.
Author Response
Reviewer Comment 1:
Reply: (1) According to the reviewer's suggestion, submit the article to a professional English editing agency to revise the abstract and the grammar, spelling, and sentence structure in the article; (2) According to the reviewer's suggestion, the ASP The definition of is supplemented on lines 34-35 of the first page; (3) according to a very short incomplete sentence on line 40 of the first page proposed by the reviewer, the author has deleted the sentence after careful inspection; (4) The description of the residual oil in the corners has been supplemented in the Introduction, and the Introduction has been completely revised; (5) The COMSOL simulation method and its importance to the research work have been included in the Introduction as suggested by the reviewers Supplemented; (6) the concept of corner-shaped residual oil and polymer flooding is described in detail by the author of the article under Section 2.6, and microscopic visualization images are used to characterize the difference between water flooding and polymer flooding; (7) According to the The Discussion section has been fully revised at the reviewer's suggestion, and the revised section is marked in blue.

Reviewer 2 Report
In order to study the microscopic production mechanism of corner residual oil after polymer flooding, microscopic visualization oil displacement technology and COMSOL finite element numerical simulation methods are used. The influence of the viscosity and interfacial tension of the oil displacement system after polymer flooding on the movement mechanism of the corner residual oil was studied.
In this work, many groups experiments are peformed with different conditions. Very interesting works.
Some minor suggestions:
- Many sentences are too long.
- Please present the highlights of the avaiable works, what is your new advances.
- In equation 3, what is the meaning of the symbol n?
- Improve the writting English.
- Improve the figures.
Author Response
Reviewer Comment 2:
Reply 1: According to the reviewer's suggestion, the author of this article has sought the help of a professional English editing agency to revise the article comprehensively. The following is the proof.
Reply 2: There are three main innovations in this paper: (1) Microscopic visualization oil displacement experiments of oil displacement systems with different viscosities and interfacial tensions after polymer flooding are carried out by using the microscopic visualization model of corner glass etching. The dynamic image of the residual oil starting to migrate, analyzes the influence of viscosity and interfacial tension on the corner-shaped residual oil; (2) Through the force analysis and theoretical calculation of the corner-shaped residual oil with water-wet properties, the wettability is clarified On the influence of the corner-shaped residual oil production, the conditions of the residual oil in the production water-wet property corner were quantitatively analyzed: the viscosity of the displacement fluid or the seepage velocity increased by 100-1000 times; (3) According to the distribution of the residual oil in the corner-shaped residual oil The finite element numerical simulation software COMSOL is used to establish a model to define the displacing phase and the displaced phase fluid. According to the conditions of different viscosities and interfacial tensions, the remaining oil saturation, velocity and flow field, velocity and flow field during the displacement process are simulated. Changes in parameters such as pressure flow field, displacement streamline, etc., analyze the producing mechanism of corner-shaped residual oil.
The new progress of this paper is to combine the microscopic visualization oil displacement technology with the finite element numerical simulation technology, to carry out a detailed analysis of the corner-shaped microscopic residual oil from both physical and numerical simulations, and focus on the analysis of the displacement phase viscosity, interfacial tension And the effect of wettability on the production mechanism of residual oil in the corner cube.
Reply 3: In Equation 3, the symbol n represents the power-law exponent of the polymer solution, and there is a note on line 212, which is uniformly valued at 0.5 in this paper.
Reply 4: The English grammar, spelling, sentence pattern and other issues have been comprehensively revised by a professional English editing agency, which has been explained above.
Reply 5: Figure 10 and Figure 13 in this article have been redrawn using Origin software, as shown below.
Figure 10. Saturation variation of corner residual oil under different viscosity conditions.
Figure 13. Variation of corner oil saturation under different interfacial tension.

Reviewer 3 Report
Generally, the work were well designed and conducted, and I would like to suggest that the manuscript could be accepted after minor revision.
Q1, This paper does not seem to analyze the influence of polymer solutions of different molecular weights on the research. What is the point of using polymers of different molecular weights to study?
Q2, Line 89 should has a reference, but it's missed.
Author Response
Reviewer Comment 3:
Reply 1: In this article, the 12 million molecular weight polymer is used during conventional polymer flooding on site, while the 25 million molecular weight polymer is the ultra-high molecular weight polymer used after conventional polymer flooding. In this study, only the difference in solution viscosity is used. To distinguish the two displacement methods, the effect of polymer molecular weight on oil displacement efficiency is not described in detail.
Reply 2: Line 89 is indeed missing references, 32nd and 33rd references have been added. As follows:
The purpose of this work is to use the combination of macroscopic oil displacement experiment and microscopic visualization oil displacement technology to explore the microscopic plugging control mechanism of the new heterogeneous combined oil displacement system on the remaining oil after polymer flooding[32-33]。

Round 2
Reviewer 1 Report
The manuscript has been substantially improved.
Manuscript includes the following two sections.
3- Results and Discussion
4- Discussion
I don't think that section 4- is necessary to be a separate section. It may included as part of either in the Results and Discussion section or the Summary and Conclusion section.
The manuscript can be published in Polymer after this minor modification.
Author Response

(The authors gave the same response as above.)
